# Low-Temperature Rapid Sintering of Dense Cubic Phase ZrW_2−*x*_Mo*_x_*O_8_ Ceramics by Spark Plasma Sintering and Evaluation of Its Thermal Properties

**DOI:** 10.3390/ma15134650

**Published:** 2022-07-01

**Authors:** Hui Wei, Jian Mei, Yan Xu, Xu Zhang, Jing Li, Xiaoyong Xu, Yang Zhang, Xiaodong Wang, Mingling Li

**Affiliations:** 1Key Laboratory of Novel Ceramic and Powder Engineering, Chaohu University, Hefei 238024, China; 053081@chu.edu.cn (J.L.); xxy1310@163.com (X.X.); zhangyang_sut@163.com (Y.Z.); wxd551@163.com (X.W.); 053014@chu.edu.cn (M.L.); 2Baoji Oilfield Machinery Co., Ltd., Baoji 721002, China; meijian999@cnpc.com.cn; 3Hefei Jingchuang Technology Co., Ltd., Hefei 231500, China; xy87325153@163.com (Y.X.); zhangxu05413@163.com (X.Z.)

**Keywords:** dense ZrW_2−*x*_Mo*_x_*O_8_ bulk ceramics, sol–gel-hydrothermal method, spark plasma sintering, negative thermal expansion, tunable thermal expansion

## Abstract

In this study, we report a low-temperature approach involving a combination of a sol–gel hydrothermal method and spark plasma sintering (SPS) for the fabrication of cubic phase ZrW_2−x_Mo_x_O_8_ (0.00 ≤ *x* ≤ 2.00) bulk ceramics. The cubic-ZrW_2−x_Mo_x_O_8_ (0.00 ≤ *x* ≤ 1.50) bulk ceramics were successfully synthesized within a temperature range of 623–923 K in a very short amount of time (6–7 min), which is several hundred degrees lower than the typical solid-state approach. Meanwhile, scanning electron microscopy and density measurements revealed that the cubic-ZrW_2−x_Mo_x_O_8_ (0.00 ≤ *x* ≤ 1.50) bulk ceramics were densified to more than 90%. X-ray diffraction (XRD) results revealed that the cubic phase ZrW_2−*x*_Mo*_x_*O_8_ (0.00 ≤ *x* ≤ 1.5) bulk ceramics, as well as the sol–gel-hydrothermally synthesized ZrW_2−*x*_Mo*_x_*O_7_(OH)_2_·2H_2_O precursors correspond to their respective pure single phases. The bulk ceramics demonstrated negative thermal expansion characteristics, and the coefficients of negative thermal expansion were shown to be tunable in cubic-ZrW_2−*x*_Mo*_x_*O_8_ bulk ceramics with respect to *x* value and sintering temperature. The cubic-ZrW_2−*x*_Mo*_x_*O_8_ solid solution can thus have potential applications in electronic devices such as heat sinks that require regulation of thermal expansion.

## 1. Introduction

Cubic phase ZrW_2−*x*_Mo*_x_*O_8_ (cubic-ZrW_2−*x*_Mo*_x_*O_8_) is a well-known substitutional solid solution material with unique negative thermal expansion (NTE) properties. It has been extensively considered for use in some applications such as precision mechanical engineering and optics that require precisely controllable thermal expansion [1,2,3,4,5,6,7,8,9]. Although some compounds (e.g., ice [10], SiO_2_ [11], PbTiO_3_ [12]) are known to exhibit anisotropic NTE over a limited temperature range. The cubic-ZrW_2−*x*_Mo*_x_*O_8_ shows a three-dimensional isotropic NTE coefficient over a wide temperature range, which is considered to be an ideal filler to improve the thermal expansion coefficient in composites [13,14,15,16,17,18]. Among this substitutional solid solution, the NTE coefficients of cubic-ZrW_2−*x*_Mo*_x_*O_8_ (*x* = 0.0, 1.0, 2.0) have been extensively investigated. The most widely accepted value is −4.8 ~ −8.8 × 10^−6^ K^−1^ for cubic-ZrW_2_O_8_, −4.5 ~ −8.7 × 10^−6^ K^−1^ for cubic-ZrWMoO_8_, and −5.0 ~ −8.0 × 10^−6^ K^−1^ for cubic-ZrMo_2_O_8_, which has been reported by Evans et. al. [19,20,21,22,23]. The discontinuous change in the NTE coefficients of each material is related to a reversible order-disorder phase transformation from a low-temperature acentric cubic phase (α-ZrW_2−*x*_Mo*_x_*O_8_, space group P 2_1_ 3) to a high-temperature centric cubic phase (β-ZrW_2−*x*_Mo*_x_*O_8_, space group P a $\bar{3}$) [1,2,3,4,5,6,7,13,14,24]. Interestingly, this phase transformation temperature decreases with the increase in Mo content: from 443 K (*x* = 0.0) to 270 K (*x* = 1.0) and then dropping to a value of 200 K (*x* = 2.0) [19,20,21,22,23]. Another polymorph, trigonal phase γ-ZrW_2−*x*_Mo*_x_*O_8_ (1.0 ≤ *x* ≤ 2.0) with a large positive thermal expansion coefficient of 23 × 10^−6^ K^−1^, appears when the high-temperature cubic phase is heated further [4,5,6,7,25]. To the best of our knowledge, the rapid change of the thermal expansion coefficient due to the cubic-trigonal phase transformation is detrimental to practical applications. To obtain pure cubic-ZrW_2−*x*_Mo*_x_*O_8_, it is necessary to have an accurate knowledge of the temperature interval of the cubic phase with different *x* values.

Similar to many hard-to-sinter ceramics, the most challenging problem for Mo-doped ZrW_2_O_8_ substitutional solid solution is to devise a method to sinter the dense bulk ceramics. The phase diagram of the ZrO_2_-WO_3_ system indicates that ZrW_2_O_8_ is thermodynamically stable over a limited temperature range; precisely from 1380 K to 1520 K, while it is metastable from absolute 0 K to 1050 K. As it slowly cools from its stable state, ZrW_2_O_8_ will decompose into ZrO_2_-ss and WO_3_. Hence, it is difficult to generate the pure phase via the high-temperature solid-state route [26,27,28]. Thus, the investigation of alternative low-temperature liquid-phase methods is rather mandatory. It has been reported that ZrW_2_O_8_ powders can be prepared by dehydrating its hydrate precursor ZrW_2_O_7_(OH)_2_·2H_2_O, which can be obtained using liquid-phase methods such as sol–gel [1,2,29,30,31], hydrothermal [3,5,32,33,34,35,36] and co-precipitation [35]. Kanamori et. al. [32] first reported that the cubic phase undoped ZrW_2_O_8_ bulk ceramic, with a relative density of 98.6%, could be prepared by using the combined sol–gel and a spark plasma sintering (SPS) technique at a low temperature; precisely (873 K). There are, however, no reports about the fabrication of dense cubic-ZrW_2−*x*_Mo*_x_*O_8_ ceramics.

In the present study, a sol–gel-hydrothermal method was used for synthesizing ZrW_2−*x*_Mo*_x_*O_7_(OH)_2_·2H_2_O (0.00 ≤ *x* ≤ 2.00) precursors. Subsequently, the cubic-ZrW_2−*x*_Mo*_x_*O_8_ ceramics were prepared by SPS using pre-calcination powders as a raw material. Thermogravimetric-differential thermal analyzer and high-temperature X-ray diffraction techniques were employed to investigate the sintering temperatures of cubic-ZrW_2−*x*_Mo*_x_*O_8_ ceramics. In addition, the NTE coefficients of cubic-ZrW_2−*x*_Mo*_x_*O_8_ bulk ceramics were measured using a thermodynamic analyzer. The impact of various sintering temperatures on the relative densities and NTE coefficients of cubic-ZrW_2−*x*_Mo*_x_*O_8_ bulk ceramics was also investigated.

## 2. Materials and Methods

### 2.1. Synthesis of ZrW_2−x_Mo_x_O_7_(OH)_2_·2H_2_O Precursors

ZrW_2−*x*_Mo*_x_*O_7_(OH)_2_·2H_2_O was prepared by a sol–gel-hydrothermal method. In the first stage (sol–gel stage), ZrOCl_2_·8H_2_O (Kishida Chemical, Osaka, Japan, 99%) was dissolved in 4 mol·L^−1^ acetic acid and 2-butanol in the air for 3 h. At the same time, WCl_6_ (High Purity Chemicals, Saitama, Japan, 99.99%) and MoCl_5_ (Sigma Aldrich, Saint Louis, MO, USA, 95%) were separately dissolved in ethanol in a nitrogen atmosphere in a stoichiometric ratio of W:Mo = (2 − *x*): *x* (*x* = 0.0, 0.1, 0.2, 0.5, 0.6, 0.7, 1.0, 1.5, 2.0). Following 3 h of stirring, the tungsten and molybdenum solutions were poured into the zirconium solution and allowed to stir for 72 h. The mixed solution comprising tungsten, molybdenum, and zirconium ions was heated in a silicone oil bath at around 360 K. Sol–gel precursors were obtained after the solution had been completely evaporated. The resultant sol–gel precursors were dissolved in 60 mL of distilled water during the second stage (hydrothermal stage). It was then added to a 100 mL Teflon-lined Parr bomb placed in a mantle heater at 453 K and allowed to stir continuously for 18 h. The sol–gel-hydrothermal precursors were subsequently centrifuged, washed with distilled H_2_O, and dried in an oven at 333 K.

### 2.2. Fabrication of Cubic-ZrW_2−x_Mo_x_O_8_ Ceramics

In this study, an SPS apparatus (SPS-515S; Fuji Electronic Industrial, Saitama, Japan), with a pulse duration of 3.3 ms and on/off pulse intervals of 12:2 s, was applied for the preparation of dense cubic-ZrW_2−*x*_Mo*_x_*O_8_ bulk ceramics. The sol–gel-hydrothermal precursors were pre-calcined at 573–723 K in a drying oven (DX-41, Yamato Scientific, Tokyo, Japan). The pre-calcination powders were ground and sieved to 45–100 μm (typical batch of about 1.0 g), and then placed into a graphite die (10 mm in diameter and 20 mm in depth) with a layer of protective carbon sheet. The sintering temperature was set to the temperatures based on the high-temperature XRD results of the ZrW_2−*x*_Mo*_x_*O_7_(OH)_2_·_2_H_2_O precursors. Each sample’s heating rate and holding period were 100 K·min^−1^ and 10 min, respectively. During the SPS procedure, the applied pressure of 50 MPa was held constant in an argon atmosphere until the conclusion of the sintering period. The volume contraction between the two graphite punches, namely the *Z*-axis linear shrinkage value, was recorded every 30 s during SPS.

### 2.3. Evaluation Method

The phase identification of the ZrW_2−*x*_Mo*_x_*O_7_(OH)_2_·2H_2_O precursors and cubic-ZrW_2−*x*_Mo*_x_*O_8_ ceramics was carried out using X-ray powder diffraction (XRD; Rigaku, Ultima IV, Tokyo, Japan). The XRD data were collected at a scan speed of 6°·min^−1^ in the 2θ range from 10° to 40° under 40 kV/40 mA with Cu-Kα radiation (λ = 0.15406 nm) using a continuum scanning method. To investigate the temperature range in which -ZrW_2−*x*_Mo*_x_*O_8_ exists as a cubic structure, thermogravimetric-differential thermal analyses (TG-DTA; SHIMADZU, DTG-60/60H, Kyoto, Japan) and high-temperature X-ray diffraction (HT-XRD; Empyrean, PANalytical, EA Almelo, The Netherlands) were utilized. The TG-DTA experiments were carried out with airflow in the range between room temperature and 1073 K, at a heating rate of 10 K·min^−1^. The HT-XRD data for ZrW_2−*x*_Mo*_x_*O_7_(OH)_2_·2H_2_O precursors were recorded at the temperature ranging from room temperature to 973 K in the 2θ range of 10° to 40°. Prior to the collection of XRD data, the samples were heated to the desired temperature at a rate of 10 K·min^−1^ and kept for 2 min to equilibrate. The relative densities of the ZrW_2−*x*_Mo*_x_*O_8_ bulk ceramics were measured using Archimedes’ principle in distilled water. The density of distilled water is 0.99754 g·cm^−3^ at a room temperature of 296 K and the theoretical density of ZrW_2_O_8_ has been reported to be 5.08 g·cm^−3^ [13,14]. The microstructural analysis of the fractured surface of each bulk ceramic was performed with a scanning electron microscope (SEM; JEOL, JSM-IT100, Tokyo, Japan) along with energy dispersive spectroscopic (EDS) mapping. A thermodynamic analyzer (TMA; Shimadzu, TMA-60/60H, Kyoto, Japan), in the temperature range of 323–673 K with a heating rate of 1 K·min^−1^ was used to measure the changes in length relative to temperature.

## 3. Results and Discussion

### 3.1. Preparation of ZrW_2−x_Mo_x_O_7_(OH)_2_·2H_2_O Precursors

Figure 1 shows the XRD patterns of the ZrW_2−*x*_Mo*_x_*O_7_(OH)_2_·2H_2_O (*x* = 0.0, 0.5, 1.0, 1.5, 2.0) precursors after the (a) sol–gel stage and (b) sol–gel-hydrothermal stage. It was found that the undoped ZrW_2_O_7_(OH)_2_·2H_2_O precursor could be successfully prepared using the sol–gel method, which is in agreement with results from many past studies [1,2,29,30,31]. However, the limitations of this method are reflected in the formation of Mo-doped solid solutions. It can be observed that the crystallinity of the precursor of *x* = 0.5 is inferior to *x* = 0.0, while no specific XRD crystallization reflections can be observed in cases when *x* is greater than 1.0. By comparison, all the precursors obtained from the sol–gel-hydrothermal stage were indexed to a similar structure of ZrW_2_O_7_(OH)_2_·2H_2_O (JCPDS file no. 28-1500) or ZrMo_2_O_7_(OH)_2_·2H_2_O (JCPDS file no. 27-0994). The changes in the ratio of the relative intensities of the four strong reflections (200, 202, 310, and 312) of *x* = 0.0 and *x* = 2.0 samples may be arising from the difference in atomic numbers between Mo (42) and W (74), resulting in a difference of X-ray scattering cross-section.

The ZrW_2−*x*_Mo*_x_*O_7_(OH)_2_·_2_H_2_O precursors were measured using TG-DTA to understand better the compositional and weight variations of ZrW_2−*x*_Mo*_x_*O_8_ bulk ceramics during heating. TG-DTA curves of ZrW_2−*x*_Mo*_x_*O_7_(OH)_2_·2H_2_O (*x* = 0.0, 1.0) precursors are depicted in Figure 2a,b. Up to 500 K, the behaviors of all samples are more or less similar. An endothermic peak in the DTA accounted for 7–10% of the initial weight loss in the TG curves. This may occur due to the dehydration and dehydrogenation of the precursors. This behavior was further verified by the observation of the FT-IR spectra of ZrW_2−*x*_Mo*_x_*O_7_(OH)_2_·2H_2_O precursors reported in the literature [3,14,37]. The DTA curves of the *x* = 1.0 have a pair of exothermic peaks located near 669 K and 817 K, while only one exothermic peak is observed at 863 K for non-doped ZrW_2_O_8_. The first exothermic peak is assigned to the formation of the cubic phase and the second exothermic peak of *x* = 1.0 sample is identified as the cubic-trigonal phase transformation. It has been reported that due to the instability of cubic phase ZrW_2−*x*_Mo*_x_*O_8_ (*x* ≥ 1.0), a mixed-phase (cubic + trigonal) is produced during the conversion of the cubic phase to the trigonal phase [4,5,6,7,25]. The occurrence of this phenomenon implies that the preparation of pure cubic phase samples requires accurate analysis.

The compositional and thermal dependence of the phase behaviors of ZrW_2−*x*_Mo*_x_*O_7_(OH)_2_·2H_2_O precursors were examined by HT-XRD measurements. These measurements are anticipated to indicate the underlying mechanism responsible for the phase transition detected by the TG-DTA measurements. Figure 3a,b show the HT-XRD patterns of the ZrW_2−*x*_Mo*_x_*O_7_(OH)_2_·2H_2_O (*x* = 0.0, 1.0) precursors. It was found that, from room temperature to 373 K, the two precursors maintained their corresponding precursor phases. At 423 K, the ZrW_2_O_7_(OH)_2_·2H_2_O precursors became amorphous, then transformed into the cubic phase ZrW_2_O_8_ at 823 K and a certain amount of WO_3_ decomposed at 903 K. The ZrWMoO_7_(OH)_2_·2H_2_O precursors, on the other hand, changed into an orthorhombic phase, then transformed into cubic-ZrWMoO_8_ at 753 K, and ultimately converted into a cubic + trigonal mixed-phase after 833 K. This observation indicates that the change in crystal structure during heating depends on the Mo concentration and synthesis temperature of the cubic phase. The HT-XRD results shown in Figure 4 shows the overall process of phase transformation for the ZrW_2−*x*_Mo*_x_*O_7_(OH)_2_·2H_2_O (*x* = 0.0, 0.1, 0.2, 0.5, 0.6, 0.7, 1.0, 1.5, 2.0) system during the heating procedure. The synthesis temperature of cubic-ZrW_2–*x*_Mo*_x_*O_8_ and the width of the synthetic “window” for the cubic phase decreases as the *x* value increases: a gradual shift from the interval of 80 K between 823–903 K (*x* = 0.0) to an interval of 50 K between 753–803 K (*x* = 1.0) and fell to the narrow range only at 623 K for *x* = 2.0. In this study, the sintering temperatures were set to 923 K and 873 K for *x* = 0.0, *x* = 0.1, *x* = 0.2; 873 K and 823 K for *x* = 0.5*, x* = 0.6*, x* = 0.7; 773 K and 823 K for *x* = 1.0; 743 K and 723 K for *x* = 1.5; 623 K for *x* = 2.0, respectively.

### 3.2. Fabrication of Dense Cubic-ZrW_2–x_Mo_x_O_8_ Bulk Ceramics

Figure 5 shows the *Z*-axis linear shrinkage curve of the ZrW_2_O_8_ bulk ceramic at a sintering temperature of 923 K. In the temperature range of 300–430 K, a slight *Z*-axis shrinkage is observed, which may be due to the evaporation of H_2_O and CO_3_^2−^ on the surface of the pre-calcining powders. The remarkable shrinkage caused by the densification takes place at a temperature of 450 K and ends at a set temperature of 923 K. It means that the ZrW_2_O_8_ bulk ceramics sintered at a higher temperature (923 K) are denser than those sintered at lower temperatures (873 K), where the densification change can be confirmed by SEM micrographs of the fracture surfaces of ZrW_2_O_8_ bulk ceramic shown in Figure 6. A relatively loose microstructure consisting of some pores was observed in the 873 K-sintered ceramic (Figure 6a). In comparison, the 923 K-sintered ceramic was dense due to the decrease in the number of pores (Figure 6b). In addition, the *Z*-axis linear shrinkage curve at the time of holding temperature does not affect the compactness of the bulk ceramic, which means that it takes only 6–7 min to be prepared using the SPS technique. Figure 7 shows the relative densities of ZrW_2−*x*_Mo*_x_*O_8_ bulk ceramics prepared by SPS at various sintering temperatures. It was found that the relative density of ZrW_2−*x*_Mo*_x_*O_8_ bulk ceramics increased with the rise in sintering temperature, which is consistent with the results of the *Z*-axis linear shrinkage curve. The relative densities of cubic-ZrW_2−*x*_Mo*_x_*O_8_ bulk ceramics were 96.5% and 91.1% for the *x* = 0.0 sample sintered at 923 K and 873 K, respectively; 95.3% and 93.6% for the *x* = 0.5 sample sintered at 873 K and 823 K respectively; 96.3% and 91.2% for the *x* = 1.0 sample sintered at 823 K and 773 K respectively; and near 90% for the *x* = 1.5 sample sintered at 743 K. This observation implies that SPS can densely fabricate the hard-to-sinter ceramics. On the other side, the relative density of *x* = 1.5 sintered at 723 K and *x* = 2.0 samples reached approximately 80%, which can be attributed to its extremely low sintering temperature. Figure 6c–f shows the SEM micrographs of the fractured surface of ZrW_2−*x*_Mo*_x_*O_8_ (*x* = 0.5, 1.0, 1.5, 2.0) bulk ceramics. The *x* ≤ 1.5 bulk ceramics are evidently dense, while the fracture surface of the *x* = 2.0 sample exhibits many granular aggregates that are not tightly bound together. Figure 6g presents the SEM micrographs on the polished surface of ZrW_2_O_8_ ceramic and the corresponding EDS mapping of Zr, W, and O elements. No element degradation was identified from the EDS mapping, confirming that a single phase is present in the ZrW_2_O_8_ sample.

Figure 8 shows the XRD patterns of the cubic-ZrW_2−*x*_Mo*_x_*O_8_ bulk ceramics at sintering temperatures of 923 K (*x* = 0.0), 873 K (*x* = 0.5), 823 K (*x* = 1.0), 743 K (*x* = 1.5), and 623 K (*x* = 2.0). The profiles of solid solution ceramics (*x* ≤ 1.5) were identified as those of a single phase of cubic-ZrW_2_O_8_ (JCPDS file no. 50-1868), while the ZrMo_2_O_8_ bulk ceramic was found to contain a small amount of trigonal phase, indicating that the synthesis of a single-phase α-ZrMo_2_O_8_ is rather difficult. The absence of some special reflections (221, 310) for *x* = 1.0 and *x* = 1.5 samples is related to the occurrence of an order-disorder phase transformation, which shows variation in the space group from the low-temperature phase (space group: P 21 3) to the high-temperature phase (space group: P a $\bar{3})$. The most important finding of the current study is that low temperature is a requirement for sintering pure dense cubic ZrW_2−*x*_Mo*_x_*O_8_ bulk ceramics. In general, the fabrication of densified ZrW_2_O_8_ polycrystals requires a very high sintering temperature of more than 1373 K using the solid-phase method. The current study revealed that the SPS technique could reduce the sintering temperature required for high densification to 773–923 K. This low-temperature densification is associated with sintering compression pressure and the SPS pulsed direct current. Figure 9 shows the calculated and theoretical values of the lattice parameters of cubic-ZrW_2−*x*_Mo*_x_*O_8_ bulk ceramics according to Vegard’s law. It can be seen that the lattice parameters of all solid solution intervals decrease linearly with the increase of Mo content. Moreover, this linear decreasing trend of the lattice parameters is in general agreement with Vegard’s law, which indicates that the Mo^6+^ are a solid solution substituted to the W site. In addition, the lowering of the lattice parameters due to the addition of Mo is expected to induce distortion of the crystal structure or decrease the relaxation area due to lattice volume contraction, which may affect the thermal expansion and phase transition behavior.

### 3.3. Evaluation of NTE Coefficients of Dense Cubic-ZrW_2−x_Mo_x_O_8_ Bulk Ceramics

The relative thermal expansion of cubic-ZrW_2−*x*_Mo*_x_*O_8_ dense ceramics was measured by thermo-mechanical analyzer equipment. Figure 10a,b shows the change in relative thermal expansion as a function of temperature for the samples (*x* = 0.0, 0.5, 1.0, 1.5) sintered at higher temperatures and lower temperatures, respectively. It was evident that the thermal expansion curves decreased with increasing temperature, which indicates the NTE characteristics of the obtained cubic-ZrW_2−*x*_Mo*_x_*O_8_ ceramics. A change in the NTE coefficient, which corresponds to the phase transformation, is attributed to the rapid shift in the slope of *x* = 0.0 and *x* = 0.5 samples. The phase transformation temperature shifts to lower temperatures, from about 440 K (*x* = 0.0) to 350 K (*x* = 0.5), and drops to below room temperature when *x* ≥ 1.0, which is consistent with many previous studies conducted on ZrW_2−*x*_Mo*_x_*O_8_ powders. The repeated changes between positive and negative of *x* = 1.5 (sintered at 723 K) thermal expansion curves may be related to the sample with a low relative density.

It should also be noted that the Mo substitution does not only cause a decrease in phase transformation temperature and changes the NTE coefficients. The NTE coefficients calculated from the change in dimensions of material concerning temperature have been shown in Table 1. For the non-doped ZrW_2_O_8_ sample (sintered at 923 K), the NTE coefficients are estimated to be −8.68 × 10^−6^ K^−1^ for the α phase and −4.61 × 10^−6^ K^−1^ for the β phase. The measured coefficients have values similar to the values reported for ZrW_2_O_8_ powders in the literature (α-ZrW_2_O_8_: −8.8 × 10^−6^ K^−1^; β-ZrW_2_O_8_: −4.9 × 10^−6^ K^−1^). In addition, the ZrW_2_O_8_ sample (sintered at 873 K) with slightly smaller NTE coefficients (−8.44 × 10^−6^ K^−1^ for α phase and −4.05 × 10^−6^ K^−1^ for β phase) may be related to the pores in the bulk ceramic. The majority of cubic-ZrW_2−*x*_Mo*_x_*O_8_ bulk ceramics exhibit similar behavior. Additionally, when the Mo content rises, the absolute value of the NTE coefficients of the phase gradually decreases. In contrast, the NTE coefficients of the β phase remain almost unchanged with changes in Mo doping and stay around 4.0 ppm·K^−1^ for all samples. Due to their simple processing, excellent heat conductivity, and electrical conductivity, aluminum matrix and copper matrix composites have been widely used in industrial production. Therefore, it is anticipated that the thermal expansion coefficient of the overall composite can be modified to match the ceramic substrate (CTE: 6–12 ppm) when cubic-ZrW_2−*x*_Mo*_x_*O_8_ is compounded with copper (CTE: 16.5 ppm·K^−1^) or aluminum (CTE: 23.1 ppm·K^−1^). Therefore, self-heating or heat from external temperature during equipment operation can be avoided, preventing deformation or fracture of the junction between the substrate and the composites.

## 4. Conclusions

In conclusion, the ZrW_2−*x*_Mo*_x_*O_7_(OH)_2_·_2_H_2_O precursors obtained by the sol–gel-hydrothermal technique were employed as the raw material for SPS sintering, and HT-XRD determined the phase change process and temperature interval of various x contents. Under pressurization, the SPS pressure was held at 50 MPa, and the sintering temperature was maintained at *x* = 0.0, *x* = 0.1, *x* = 0.2 at 923 K; *x* = 0.5, *x* = 0.6, *x* = 0.7 at 873 K; *x* = 1.0 at 823 K; and *x* = 1.5 at 773 K, which resulted in pure cubic phase ceramics with relative densities above 90%. The bulk ceramics demonstrate NTE characteristics, and the NTE coefficients were tunable in cubic-ZrW_2−*x*_Mo*_x_*O_8_ bulk ceramics for *x* value. As a result, cubic-ZrW_2−*x*_Mo*_x_*O_8_ with customizable NTE coefficients can operate as a thermal expansion inhibitor for composites, allowing them to match any required thermal expansion coefficient to overcome interfacial deformation or even breakage generated by inconsistent thermal expansion coefficients.

## Figures and Tables

**Figure 1 materials-15-04650-f001:**
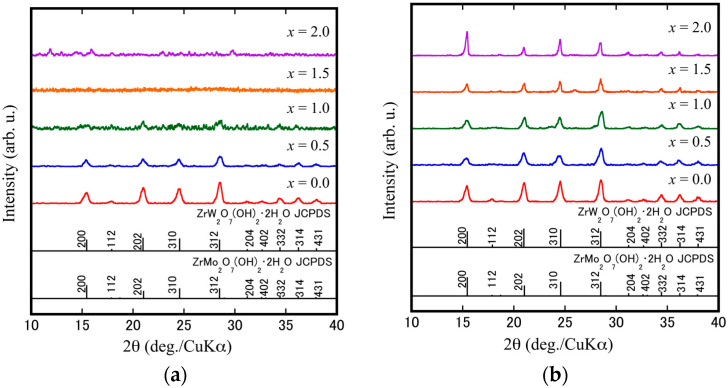
XRD patterns of the ZrW_2−*x*_Mo*_x_*O_7_(OH)_2_·2H_2_O precursors by (**a**) sol–gel stage; (**b**) sol–gel-hydrothermal stage.

**Figure 2 materials-15-04650-f002:**
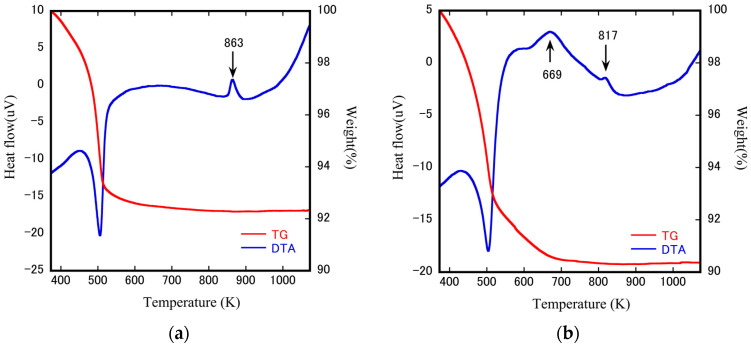
TG-DTA curves of the ZrW_2−*x*_Mo*_x_*O_7_(OH)_2_·2H_2_O (**a**) *x* = 0.0; (**b**) *x* = 1.0 precursors.

**Figure 3 materials-15-04650-f003:**
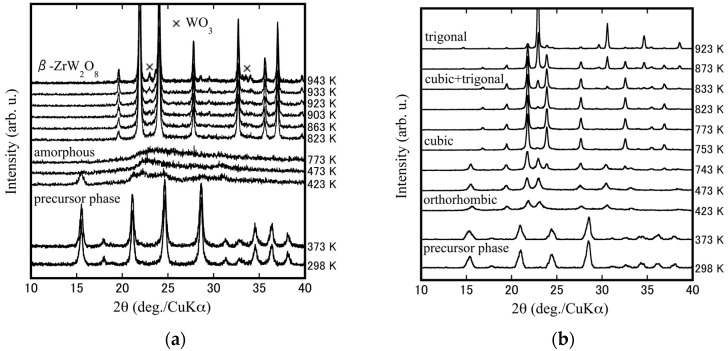
HT-XRD patterns of the ZrW_2−*x*_Mo*_x_*O_7_(OH)_2_·2H_2_O (**a**) *x* = 0.0; (**b**) *x* = 1.0 precursors.

**Figure 4 materials-15-04650-f004:**
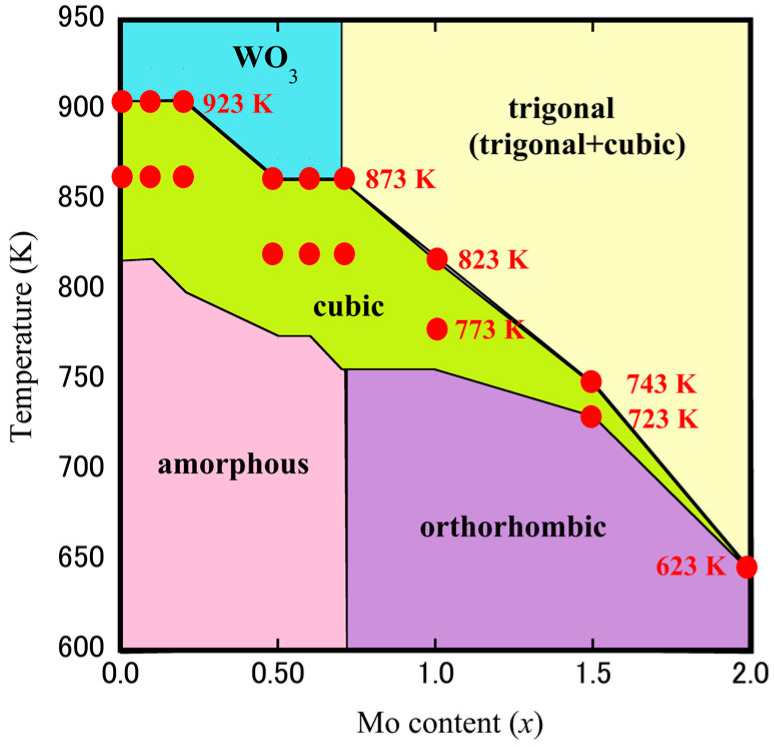
Data on the temperatures of the phase transformation for ZrW_2−*x*_Mo*_x_*O_7_(OH)_2_·2H_2_O precursors obtained by HT-XRD during the heating procedure (the red point is the setting sintering temperature during SPS of each sample).

**Figure 5 materials-15-04650-f005:**
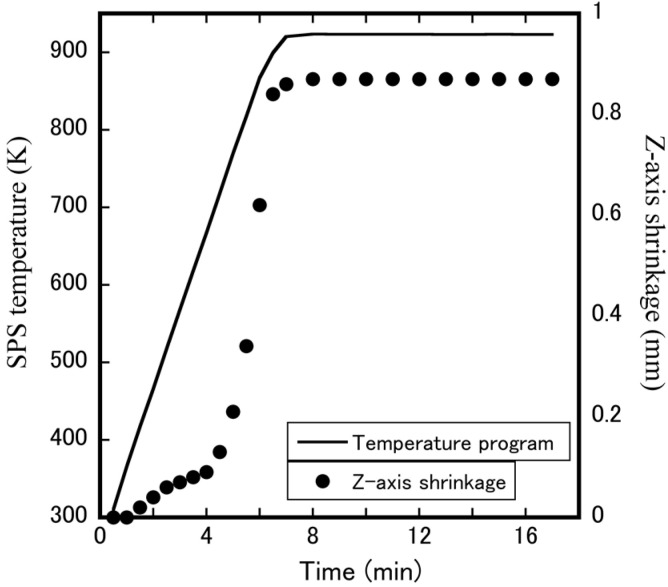
*Z*-axis linear shrinkage of pre-calcination ZrW_2_O_8_ powders during SPS.

**Figure 6 materials-15-04650-f006:**
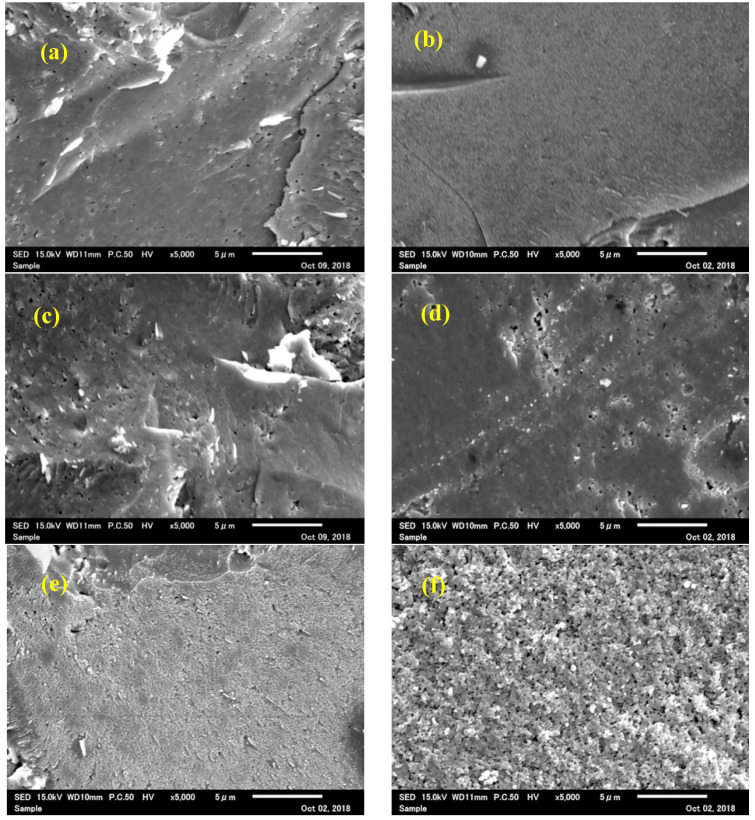
SEM micrographs of the fractured surface of cubic-ZrW_2−*x*_Mo*_x_*O_8_ ceramics (**a**) *x* = 0.0 sintered at 923 K, (**b**) *x* = 0.0 sintered at 873 K, (**c**) *x* = 0.5 sintered at 873 K, (**d**) *x* = 1.0 sintered at 823 K, (**e**) *x* = 1.5 sintered at 743 K, (**f**) *x* = 2.0 sintered at 623 K, (**g**) SEM micrograph of the polished surface of cubic-ZrW_2_O_8_ bulk ceramics with the corresponding EDS mapping of Zr, W, and O elements.

**Figure 7 materials-15-04650-f007:**
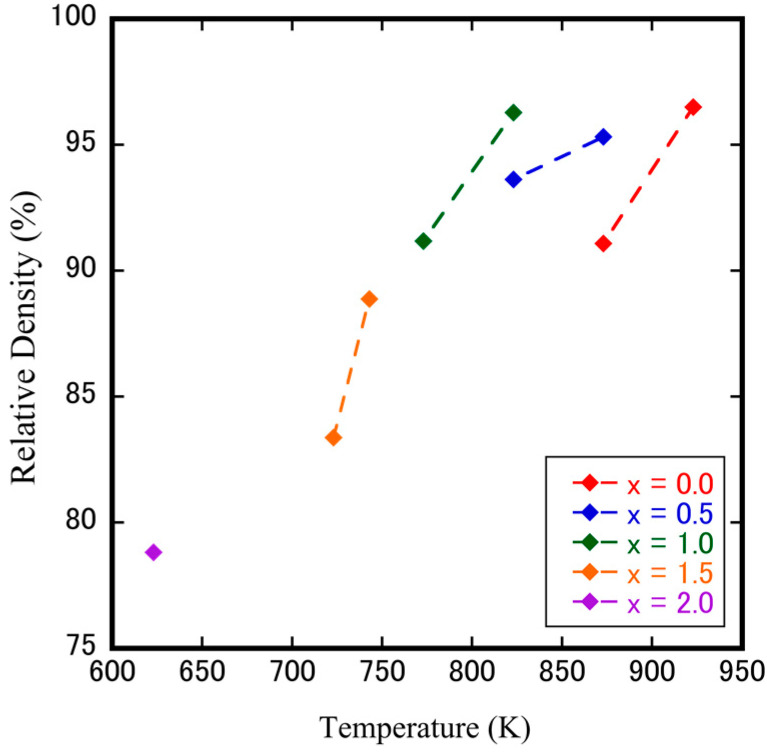
Relative densities of ZrW_2−*x*_Mo*_x_*O_8_ bulk ceramics sintered at various sintering temperature.

**Figure 8 materials-15-04650-f008:**
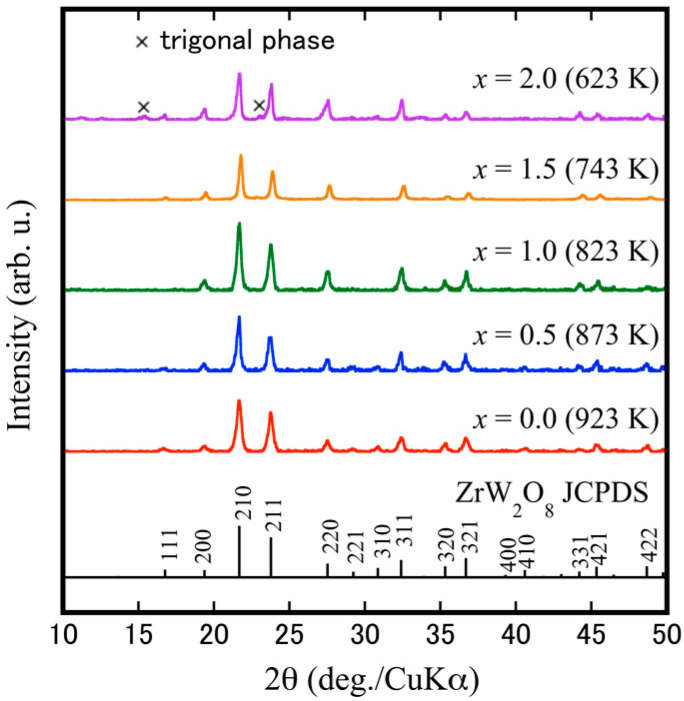
XRD results of cubic-ZrW_2−*x*_Mo*_x_*O_8_ bulk ceramics.

**Figure 9 materials-15-04650-f009:**
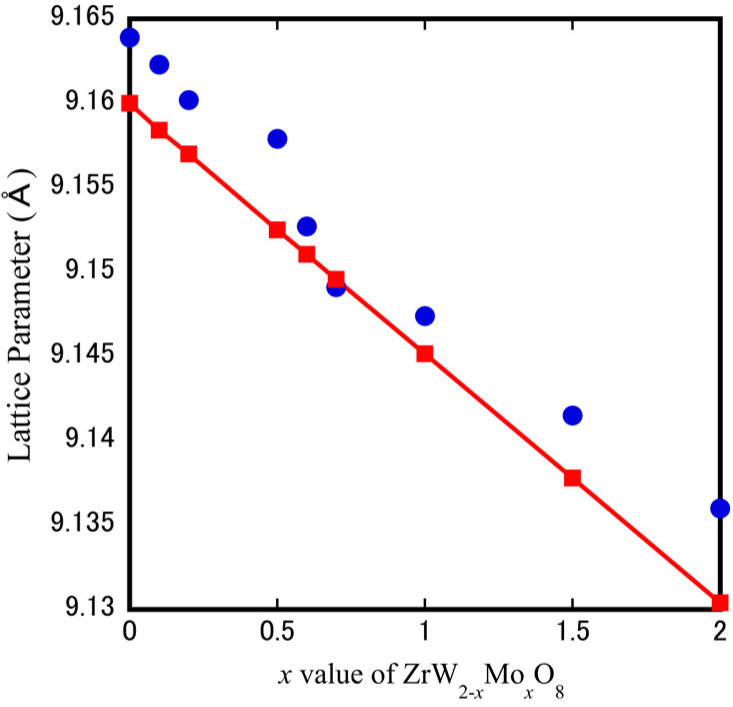
The trend of the lattice parameters of cubic-ZrW_2−*x*_Mo*_x_*O_8_ (the blue circle marks represent the calculated value of lattice parameters, and red diamond marks represent the theoretical value based on the Vegard’s law).

**Figure 10 materials-15-04650-f010:**
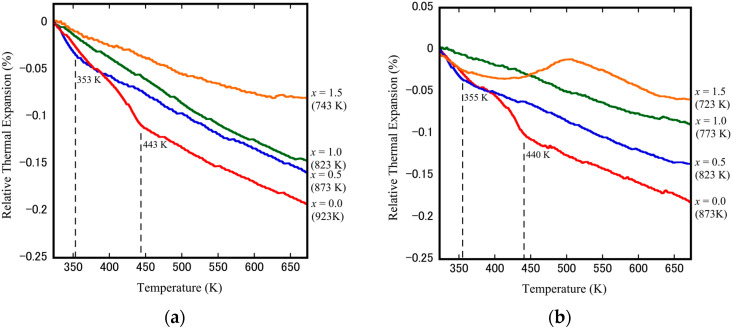
Thermomechanical analyzer results of cubic-ZrW_2−*x*_Mo*_x_*O_8_ ceramics at different sintering temperature: (**a**) the higher sintering temperature; (**b**) the lower sintering temperature.

**Table 1 materials-15-04650-t001:** NTE coefficients of cubic-ZrW_2−*x*_Mo*_x_*O_8_ bulk ceramics by TMA.

	Measured NTE Coefficients of ZrW_2−*x*_Mo*_x_*O_8_ Bulk Ceramics [×10^−6^ K^−1^]				Reported NTE Coefficients of ZrW_2−*x*_Mo*_x_*O_8_ Powders [×10^−6^ K^−1^]	
	α Phase	β Phase	α Phase	β Phase	α Phase	β Phase
*x* = 0.0	−8.68 (923 K)	−4.61 (923 K)	−8.44 (873 K)	−4.05 (873 K)	−8.80 [12]	−4.90 [12]
*x* = 0.1	−7.90 (923 K)	−3.58 (923 K)	−7.85 (873 K)	−3.22 (873 K)	—	—
*x* = 0.2	−8.26 (923 K)	−4.47 (923 K)	−7.79 (873 K)	−4.17 (873 K)	—	−3.5 [5]
*x* = 0.5	−12.3 (873 K)	−4.12 (873 K)	−10.3 (823 K)	−3.80 (823 K)	−8.79 [38]	−4.77 [38]
*x* = 0.6	−7.82 (873 K)	−4.60 (873 K)	−5.70 (823 K)	−4.01 (823 K)	—	−3.5 [5]
*x* = 0.7	—	−3.52 (873 K)	—	−2.91 (823 K)	—	—
*x* = 1.0	—	−4.28 (823 K)	—	−2.50 (773 K)	−9.00 [16]	−5.50 [16]
*x* = 1.5	—	−2.79 (773 K)	—	—	—	−3.56 [38]

## Data Availability

The data that support the findings of this study are available from the corresponding author upon reasonable request.

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
