# Peer review of "Low-Temperature Rapid Sintering of Dense Cubic Phase ZrW2−xMoxO8 Ceramics by Spark Plasma Sintering and Evaluation of Its Thermal Properties"

_materials, 2022, doi:10.3390/ma15134650_

Round 1
Reviewer 1 Report
The manuscript "Low-temperature rapid sintering of dense cubic phase ZrW2-2 xMoxO8 ceramics by spark plasma sintering and evaluation of its thermal properties" is presenting the sintering of the cubic phase by fast SPS method. The results are discussed well and are giving an additional impact on the material preparation.
The abstract could be improved by writing clearly the main reason and results of the investigation. It was misleadingly formulated that SEM and density measurement detect the cubic phase (page 1, line 17/18). This should be formulated correctly.
In the experimental part, "the mixed solution comprising tungsten, molybdenum, and zirconium" is formulated in a misleading way (page 2, line 91). Here, ions are meant but the formulation could be used for metals too. Please correct this.
The results are presented clearly and discussed scientificly. On page 4, line 159 DTA-TG is described "for investigating the temperature region in 159 which they exist as a cubic phase". With this method, it is not possible to detect phases. Only the crystallisation or transition of a phase can be measured. Therefore, the wording has to be correct here.
The conclusion can be improved. It sould be given a short summary of the results and a reflection of the results in correlation to the intension of the investigation. Therefore, the best way to prepare the cubic phase should be described.
Reviewer 2 Report
Article has been written nearly 3 years ago, there is no new references except two in 2018. Rear part of the conclusion is very confusing. please write the finding and not the results from the table without clarity.
The following improvements need to be done before consideration of final publication.
1. Kindly give the lattice parameters trend for the compositions synthesized by sol-gel method. The details will be helpful for the readers.
2. The authors should correct the figure number (5(g) to 7 (g). I think it is mistakenly written here.
3. Please give the details of NTE coefficients values. Which values of NTE coefficients are better for practical applications?
4. The authors can provide the elements compositions from the EDS if available.
5. Please check once the typo errors eg. tend is written in place of trend etc.
Round 2
Reviewer 2 Report
accept